# Chiral Liquid Crystal Microdroplets for Sensing Phospholipid Amphiphiles

**DOI:** 10.3390/bios12050313

**Published:** 2022-05-09

**Authors:** Sepideh Norouzi, Jose A. Martinez Gonzalez, Monirosadat Sadati

**Affiliations:** 1Department of Chemical Engineering, University of South Carolina, Columbia, SC 29208, USA; snorouzi@email.sc.edu; 2Facultad de Ciencias, Universidad Autónoma de San Luis Potosí, Av. Parque Chapultepec 1570, San Luis Potosí 78210 SLP, Mexico; jose.adrian.martinez@uaslp.mx

**Keywords:** chiral liquid crystal, curved confinement, biomolecules, biosensing

## Abstract

Designing simple, sensitive, fast, and inexpensive readout devices to detect biological molecules and biomarkers is crucial for early diagnosis and treatments. Here, we have studied the interaction of the chiral liquid crystal (CLC) and biomolecules at the liquid crystal (LC)-droplet interface. CLC droplets with high and low chirality were prepared using a microfluidic device. We explored the reconfiguration of the CLC molecules confined in droplets in the presence of 1,2-diauroyl-sn-glycero3-phosphatidylcholine (DLPC) phospholipid. Cross-polarized optical microscopy and spectrometry techniques were employed to monitor the effect of droplet size and DLPC concentration on the structural reorganization of the CLC molecules. Our results showed that in the presence of DLPC, the chiral LC droplets transition from planar to homeotropic ordering through a multistage molecular reorientation. However, this reconfiguration process in the low-chirality droplets happened three times faster than in high-chirality ones. Applying spectrometry and image analysis, we found that the change in the chiral droplets’ Bragg reflection can be correlated with the CLC–DLPC interactions.

## 1. Introduction

Nowadays, the noninvasive and point-of-care biosensing platforms which can endow the real-time response and low detection limit are of substantial importance for healthcare monitoring and clinical diagnosis. Liquid crystals (LCs) are one of the potent sensing moieties which endow convenient, cost-efficient, and easy-readout responsive platforms [1]. LCs are the state of matter that represents combined properties associated with ordered crystals and disordered liquids. They are composed of anisotropic molecules with various orientational and positional orderings, forming a series of mesophase structures [2,3,4,5]. The distribution of the easy axis of the anisotropic molecules relative to the introduced director field, *n*, determines the class of liquid-crystalline mesophases. Among them is the chiral liquid crystal (CLC), with a helicoidal organization, in which the bulk elastic forces drive a continuous twisting of the easy axis of anisotropic molecules relative to each other [5].

The LC interface has been shown as a sensitive reporter system to interfacial events [6,7]. Therefore, it has been extensively exploited to design sensors for detecting biochemicals and studying biological membrane functions [8,9,10,11,12,13,14]. LCs confined in spherical droplets have received particular attention due to their higher sensitivity. In droplets, the alignment of the LC molecules at the interface can be easily perturbed by subtle interfacial changes, which is due to a large and motile available interface that is not pinned to a solid surface such as a TEM grid or glass slide.

The molecular ordering transition of the nematic LC microdroplets, in response to biological molecules, has been widely explored [15,16,17]. Sivakumar et al. designed the LC microdroplet-based biosensor to differentiate bacteria and viruses based on their outer structures [18]. They showed that nematic LC droplets transition from bipolar to radial ordering when exposed to Gram-negative bacteria and enveloped viruses. However, no transition was observed in contact with Gram-positive bacteria and non-enveloped viruses. The reorganization of lipid-decorated nematic droplet interfaces has also been used to understand binding events in biological membranes. The lipid-decorated LC interfaces have been used as a simple platform to detect various biomolecules such as bacteria, toxins, and proteins based on their chemical functionality and stereochemistry [18,19,20,21,22,23]. They remain responsive to subsequent bimolecular absorption and binding events. Bao et al. studied the molecular reorientation of lipid-decorated E7 microdroplets in the presence of an antimicrobial peptide named SMP43. They reported the ordering transition from radial to planar alignment at a micromolar concentration of the antimicrobial agent [24]. These studies elucidate the high sensitivity of the nematic LC interfaces, which can be further used to decipher the underlying biomolecule’s structure–property correlation. Moreover, the detection at the LC interface can be amplified by decorating the LC-based interface with a specific paired reactive motif. Kim et al. reported the response of poly (acrylic acid-b-4-cynobiphenyl-4-oxyundecylacrylate)-decorated 5CB droplets to glucose rather than galactose with a detection limit of 0.03 mM, which is at physiologically relevant concentrations [25].

Despite extensive research on designing responsive nematic LC emulsions, the application of CLC emulsions for screening interfacial phenomena is limited to a few studies. Gollapelli et al. used CLC droplets for detecting the bile acid [26]. They reported an enhanced detection limit of CLC droplets to cholic acid and deoxycholic acid compared to nematic counterparts. Lee and coworkers studied the response of poly (acrylic acid)-b-poly(4-cyanobiphenyl-4′-oxyundecyl acrylate)-decorated CLC droplets, which were prepared by microfluidic technique [27]. Those immobilized by the glucose oxidase and cholesterol oxidase exhibited high sensitivity to glucose and cholesterol, where the enzymatic binding reaction resulted in conformational reorientation. In the same line of study, Concellon et al. investigated the reversible interaction of IgG antibodies at the CLC interface, which enabled manipulation of the chiral pitch length through the presence of a binaphthyl unit [28]. Recently, Honaker et al. have studied the interaction of CLC droplets of 15 to 90 µm diameters with sodium dodecyl sulfate, lauric acid, and DLPC. They used the ratio of color signals and time response to differentiate the effect of biomolecular targets [29]. Moreover, Paterson et al. studied the molecular reconfiguration of chiral droplets to phospholipids with different stereochemistry. They reported the movement of the defect line in chiral droplets upon interaction with phospholipids leading to transition from Franck–Pryce to nested-cup structure [30].

Although the CLC droplets exhibit novel molecular organizations and defect structures, which arise from the interplay of bulk elastic forces and surface boundary conditions [31,32,33,34,35,36], their potential for sensing diverse biomolecular structures has not been fully explored. Moreover, the reported studies are limited to the regime of low-chirality LC droplets. Here, we have presented the interaction of CLC droplets with DLPC amphiphile as a function of droplet size and amphiphile concentration for two low- and high-chirality limits. We have investigated the effect of chirality on the response time and possible molecular reorientation. The results have been compared with the detection response of nematic microdroplets. We have found a lower threshold limit of detection for CLC droplets compared to the nematic counterparts, which can be attributed to their long-range helicoidal organization. We rely on microscopic and spectroscopic readouts to discuss the LCs’ reorganizations at the microscale as a function of DLPC concentration. This study will provide insight into the modulation of the chiral ordering by interfacial binding events, which can be exploited to design biosensors with faster response time.

## 2. Materials and Methods

### 2.1. Materials

To prepare chiral LC mixtures with different levels of chirality, nematic LC (MLC2142; EMD chemical) was mixed with a chiral dopant (S811, Merck) at the concentrations of 4 wt.% and 37 wt.%. First, they were mixed in the presence of toluene as a co-solvent; afterward, toluene (Acros Organics, Newark, NJ, USA) was evaporated using a rotary evaporator (Buchi, R-210, New Castle, DE, USA). The helical twisting power (HTP) of chiral dopant is defined as $\mathrm{HTP}=\frac{1}{C\cdot P}$, where C and P are the weight concentration and pitch length, respectively. Given the helical twisting power of S-811 on MLC2142 as 10.9 µm^−1^ [37] the pitch length of low- and high-chirality mixtures were calculated to be 2.29 µm and 248 nm, respectively. Polyvinyl alcohol (PVA, 13K-23K, and 87–89% hydrolyzed; Sigma Aldrich, St. Louis, MO, USA) was used to generate the planar alignment at the interface of the droplets. The response of CLC was examined in the presence of 1,2-diauroyl-sn-glycero3-phosphatidylcholine (DLPC, Avanti Polar Lipids, Alabaster, AL, USA). Fluorescent probe 4a-diaza-*s*-indacene-3-pentanoic acid (BODIPY-FL-C5, Molecular Probes, Eugene, OR, USA) was used to label phospholipid molecules with an excitation maximum at 503 nm and emission at 512 nm. LC, 4-cyano-4’-pentylbiphenyl (5CB, Sigma Aldrich, St. Louis, MO, USA) was used to prepare the nematic droplets.

Cross-polarized optical images were obtained using a Zeiss microscope (Axioscope 5, White Plains, NY, USA) with a 50× objective lens in reflection and transmission modes. The microscope was equipped with a Zeiss camera (Axiocam 506 color, White Plains, NY, USA). The temperature of the samples was controlled by a Linkam stage (T-96). The diffused reflected spectra were measured by a fiber optic spectrometer (Flame UV-vis, Ocean Insight, Orlando, FL, USA). Fluorescent images were obtained using inverted fluorescence microscopy (EVOS FL, Waltham, MA, USA) with a 20× objective lens. The sample cell of LC droplets was made of two coverslips (22 mm × 22 mm, No.1) separated by a PDMS elastomer as a spacer. Dynamic light scattering (DLS, Malvern Instruments, Malvern, UK) was used to measure the micellar size of DLPC in water.

### 2.2. Microfluidic Fabrication and Droplet Production

We designed a glass capillary-based microfluidic device to produce CLC droplets according to Utada et al. [38]. A cylindrical capillary (OD =1 mm, WPI) was tapered by a micropipette puller (P2000, Sutter Instrument, Novato, CA, USA), then cut by a micro forge (Narishige, MF-900, Tokyo, Japan) with controlled openings. The opening of the tapered capillary was adjusted based on the desired droplet size (Appendix A). Subsequently, it was co-axially aligned with a 20-gauge blunt needle tip inside a square capillary (OD = 1.05, Harvard square tubing).

In this device, CLC was injected through a 20-gauge blunt needle tip and 0.15 wt.% PVA aqueous solution was pumped in from the square and tapered capillary interval, breaking up the continuous CLC jet into monodispersed droplets (Appendix A). Using two syringe pumps (Elite 11 series, Harvard Apparatus, Holliston, MA, USA), the flow rate of CLC and the aqueous solution was adjusted to be 0.2 µL/min and 20 µL/min, respectively. The CLC droplets of 15, 20, 30, and 50 µm were produced using this device, and the PVA solution stabilized the droplets and induced planar anchoring at their surface.

## 3. Results and Discussions

### 3.1. Dynamic Molecular Reconfiguration of High-Chirality Droplets in the Presence of DLPC

The CLC droplets with controlled diameters were produced using the microfluidic device. The planar CLC droplets in water were stabilized with PVA solution, forcing the helical axis of the chiral structure to align normal to the interface, thereby orienting the CLC molecules parallel to the interface. The planar CLC droplets manifest a rich configurational order depending on the degree of chirality. Their orientational order spans from twisted bipolar to radial spherical structure, where the defect lines approach and intertwin as the chirality increases [35]. At a high level of chirality, symmetric concentric rings form within a droplet, which results in a cross pattern under a cross-polarized microscope.

We investigated the DLPC-induced reconfiguration of the CLC microdroplets with planar surface anchoring. DLPC amphiphiles were added to the CLC microdroplets at different concentrations. After slight agitation, the mixture was injected into the sample cell, and the molecular reconfiguration was monitored over time. The dynamic of the ordering transition was monitored using a polarized optical microscope (POM) in the reflection and transmission modes as well as spectroscopy measurements. All experiments were carried out at room temperature, where the CLC droplets are in the cholesteric phase (T = 25 °C).

According to the POM images, DLPC amphiphiles at the surface of the high-chirality LC microdroplets reorganized molecular alignment from planar to homeotropic, resulting in a distinct optical appearance change (Figure 1A,B). The CLC droplets underwent multistage reconfigurations, which initiated from the interface and propagated to the bulk (Figure 1A). Since DLPC amphiphiles form vesicles above 100 nM [39], at the concentrations used in this study, they were assumed to interact with the CLC interface in the vesicle form, at which the hydrophobic tails and acyl chains interdigitated into the CLC molecules. The vesicles reoriented at the interface, and upon penetration of their acyl chains between the CLC molecules, their zwitterionic head groups packed at the interface. Previous studies have reported the transition from bipolar to radial configuration in nematic microdroplets in the presence of phospholipid amphiphiles. However, to the best of our knowledge, there is no study on the effect of chirality and curved confinement on the sensing behavior of the CLC droplets.

To confirm the presence of DLPC amphiphiles at the CLC droplet interface, the DLPC molecules were labeled by BODIPY-FL-C5 dye, and their fluorescent images were captured using fluorescence microscopy. The representative fluorescent image in Figure 1C illustrates the high concentration of DLPC molecules at the droplet interface.

We further studied the configuration-transition dynamics of monodisperse high-chirality LC microdroplets at different concentrations of DLPC: 0.05 mM, 0.1 mM, 0.5 mM, and 1 mM (Figure 2A). With increasing the DLPC concentration, the response time of the CLC droplets was significantly decreased (Figure 2A). For example, in the case of 50 µm high-chirality droplets, doubling the DLPC concentration from 0.1 mM to 1 mM reduced the response time from 18 to 5 min.

Furthermore, having precise control over the CLC droplet size, we studied the effect of curvature on the CLC response time at a constant DLPC concentration. We found that the droplet size and accessible surface can significantly affect the CLC response time. According to Lin et al. the size effect on the LC droplets’ response time can be associated with the penetration depth via t = LA^2^/4 Ds, where LA and Ds denote the equatorial length scale of penetration and diffusion constant, respectively; therefore, the larger the droplet, the longer the penetration depth and response time [22]. We further measured the minimum concentration of DLPC amphiphiles required to induce the molecular reordering for the different CLC droplet sizes. Our results suggest that 0.2 µM concentration of DLPC is sufficient to induce complete molecular reorganization in high-chirality droplets, where no CLC droplets with intermediate orientation were observed

The response time of the CLC droplets to the DLPC concentration can be correlated to the LC-phospholipid molecules’ interfacial interactions. To quantify the interfacial interactions, the changes in Bragg reflection of chiral droplets as a function of DLPC concentration were monitored over time (Figure 2B). We used a MATLAB image processing code to deconvolute the colorimetric channels into red, green, and blue. The average shift in each colorimetric channel over the reorganization intervals can be correlated to the DLPC–LC interactions at the interface. It should be mentioned that all measurements were performed under similar conditions. At 1 mM DLPC concentration, the 50 µm CLC droplets reached the equilibrium state within 5 min, while the equilibrium configuration at 0.1 mM DLPC concentration was achieved after about 18 min of incubation time. Upon planar to homeotropic reconfiguration in the presence of 1mM DLPC concentration, average shifts of 75 ± 5, 65 ± 5, and 40 ± 5 units were observed in the red, green, and blue colorimetric channels, respectively. At 0.1 mM DLPC concentration, on the other hand, a similar colorimetric shift was achieved over a longer incubation time. It is worth mentioning that the color shift in colorimetric channels was size-dependent, and different droplet sizes covered the specific region of the color gamut upon molecular reorientation (Appendix A). Moreover, the reflection spectra of the spectroscopy measurements exhibited a distinct increase in the intensity and broadening of characteristic reflection peaks (Appendix A).

### 3.2. Dynamic Molecular Reconfiguration of Low-Chirality Droplets in the Presence of DLPC

We further studied the chirality effect on the reconfiguration of the CLC droplets in the presence of DLPC amphiphiles. By mixing 4 wt.% of chiral dopant with the nematic LC, a chiral nematic LC with a pitch length of 2.29 µm was achieved. Subsequently, the microfluidic device was used to produce low-chirality microdroplets with diameters of 15 µm, 20 µm, 30 µm, and 50 µm. Under a cross-polarized microscope, the low-chirality LC droplets with planar surface anchoring appeared with well-organized concentric rings. We studied the reconfiguration of the low-chirality LC molecules for different DLPC concentrations of 0.2 µM to 1 mM (Figure 3 and Appendix A). Interestingly, at 1 mM DLPC, the low-chirality droplets exhibited multistage, size-correlated surface-pattern evolution, which was not observed in the high-chirality droplets. Upon exposure to DLPC amphiphiles, the concentric structure was disrupted on the surface of the 50 µm droplets, and the focal conic texture formed. These patterns gradually expanded over the whole surface and transitioned to larger focal conic domains with different undulating textures than the initial one (Figure 3A). However, reducing the droplet size to below 30 µm, prevented focal conic texture formation, and instead, different orientations of stripe patterns appeared on the surface (Figure 3B,C). The changes in stripe patterns in the CLC droplets have been reported by Popov et al. for a CLC flat film in contact with DLPC amphiphiles [40]. They have shown that the CLC film transitions from fingerprint texture to polygonal, spiral, and eventually fingerprints with different pitch lengths. These multistage changes in the CLC morphologies are attributed to the chiral association of DLPC amphiphiles at the CLC–DLPC interface, affecting the chiral pitch length at the droplet surface. The initial and final pitch length of the low chiral LC droplets in the presence of DLPC amphiphiles was measured to be 2.1 µm and 1.8 µm, respectively (Appendix A).

We further investigated the response time of the low-chirality droplets as a function of droplet size and DLPC concentration. Regardless of the droplet size, we found a faster response time to DLPC amphiphiles for the low-chirality droplets than the high-chirality ones (Figure 4A). For example, upon exposure to 1 mM DLPC, the 50 µm low-chirality LC droplets underwent molecular reconfiguration in about two minutes, which was three times faster than the measured value for the same size high-chirality droplets. The faster response of the low-chirality droplets to DLPC amphiphiles can be attributed to the larger pitch length, which promotes interfacial interactions. Consistent with the high-chirality droplets, the response time of the low-chirality droplets was droplet-size-independent.

The multistage molecular reconfiguration of the low-chirality droplets in response to DLPC amphiphiles facilitated studying the structural evolution and DLPC–CLC interactions. The MATLAB image processing code was employed to deconvolute the colorimetric channels of the low-chirality droplet during the orientational transition induced by DLPC amphiphiles (Figure 4B). At 0.1 mM DLPC concentration, average shifts of 30 ± 5, 5 ± 5, and 2 ± 5 were recorded for red, green, and blue channels, respectively. By increasing the DLPC concentration to 1 mM, average shifts of 32 ± 5, 2 ± 5, and 14 ± 5 were observed for red, green, and blue channels. The POM images in the reflection mode indicated a distinct redshift in the optical appearance of the low-chirality droplets upon interaction with 1 mM DLPC amphiphiles. At low DLPC concentrations, however, the color remained bluish, in which the average count of the blue pixel value was higher.

Similar to the high-chirality droplets, the minimum concentration of 0.2 µM DLPC was sufficient to reconfigure the low-chirality LC molecules. However, the change in molecular reorientation was only observed in a few small domains on the surface. Moreover, measuring the diffused reflection spectra of 50 µm low-chirality droplets in the presence of DLPC amphiphiles demonstrated a notable increase in the intensity and width of the reflected peaks as well as a slight red shift in low-chirality droplets (Appendix A).

Finally, to provide a better picture of the chirality effect on the minimum detection limit and response time of the LC droplets, we compared the response behavior of the CLC droplets with an achiral system (Figure 5). It has been shown that nematic LCs undergo bipolar to radial order transition in the presence of DLPC amphiphiles. We found that a minimum concentration of 10 mM of DLPC amphiphiles was required to achieve complete molecular reordering in the majority (>80%) of the nematic LC droplets, which was about 50 times higher than the detection limit of the chiral droplets. Although the structural reconfiguration in the nematic droplets had been detected at lower concentrations, they are mostly trapped in the intermediate molecular configuration, and complete reordering from bipolar to radial orientation required a few hours.

## 4. Conclusions

We investigated the molecular reorientation of the CLC droplets in the presence of DLPC amphiphiles as a function of DLPC concentration. To examine the effect of curvature, a microfluidic device was used to produce monodispersed, well-controlled CLC droplets. Subsequently, the CLC–DLPC interfacial interaction was studied at two low- and high-chirality levels. Our results demonstrated that the CLC–DLPC interface undergoes planar to homeotropic ordering transition through multistage molecular reorientations. However, the low-chirality droplets detected the presence of DLPC amphiphiles three times faster than their high-chirality counterparts. Furthermore, we found that the chiral droplets underwent complete reconfiguration at significantly lower concentrations of DLPC amphiphiles than the achiral LC droplets. Finally, we showed that a detailed analysis of the reflected light intensity and wavelengths could be used to assess the DLPC–CLC interaction. These findings demonstrate the great potential of the CLC droplets in developing biosensors where the response time matters.

## Figures and Tables

**Figure 1 biosensors-12-00313-f001:**
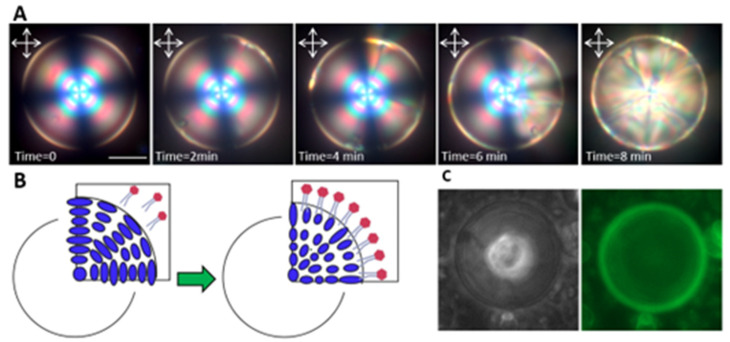
Reconfiguration of high-chirality LC droplets in the presence of DLPC amphiphiles. (**A**) Reflection mode POM images of reconfiguration dynamics in 30 µm high-chirality droplets in the presence of 0.5 mM DLPC. (**B**) Schematic of the planar to homeotropic ordering transition. (**C**) Bright-field and fluorescent image of adsorbed labeled DLPC amphiphiles on the chiral droplet’s interface (scale bar = 20 µm).

**Figure 2 biosensors-12-00313-f002:**
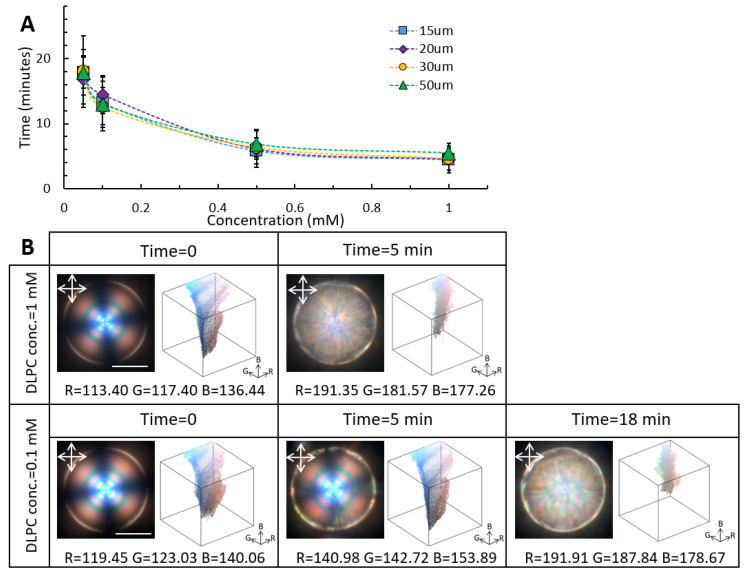
Response time of the high-chirality LC droplet as a function of droplet size and DLPC concentration. (**A**) Response time of the high-chirality droplets as a function of DLPC concentration for various droplet sizes. (**B**) Evolution of the colorimetric channels and color gamut of the 50 µm highly chiral droplets as a function of DLPC concentration (scale bar = 20 µm).

**Figure 3 biosensors-12-00313-f003:**
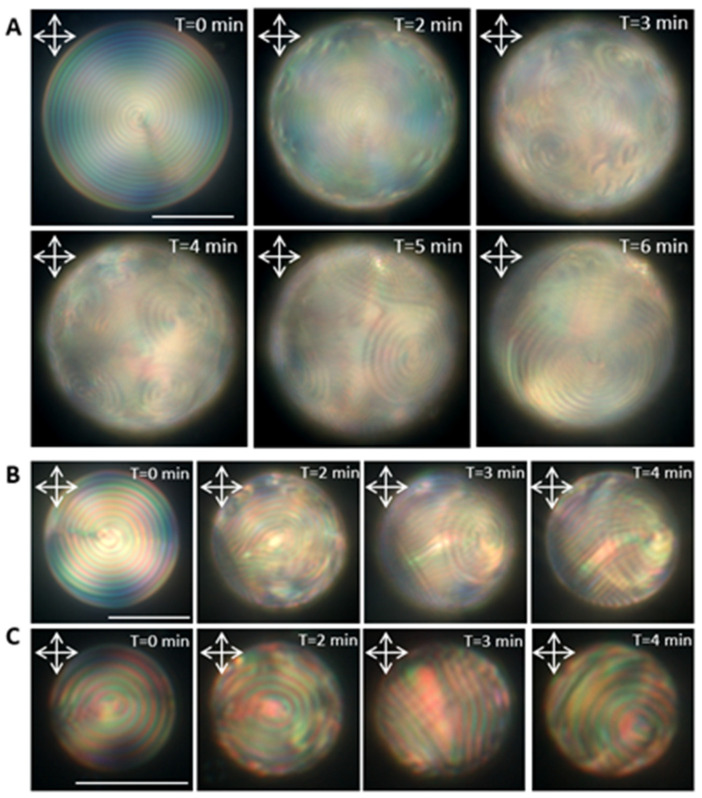
Reflection mode POM images of the reconfiguration dynamics of the low-chirality droplets of different sizes in the presence of 1 mM DLPC. (**A**) 50 µm, (**B**) 30 µm, and (**C**) 20 µm (scale bar = 20 µm).

**Figure 4 biosensors-12-00313-f004:**
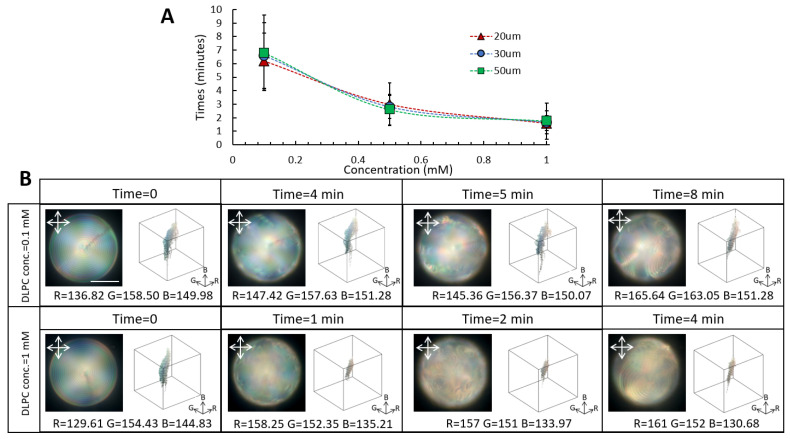
Response time of the low-chirality droplets as a function of droplet size and DLPC concentration. (**A**) Response time of the low-chirality droplets as a function of DLPC concentration for various droplet sizes. (**B**) Evolution of the colorimetric channels and color gamut of the 50 µm low-chirality droplets as a function of DLPC concentrations (scale bar = 20 µm).

**Figure 5 biosensors-12-00313-f005:**
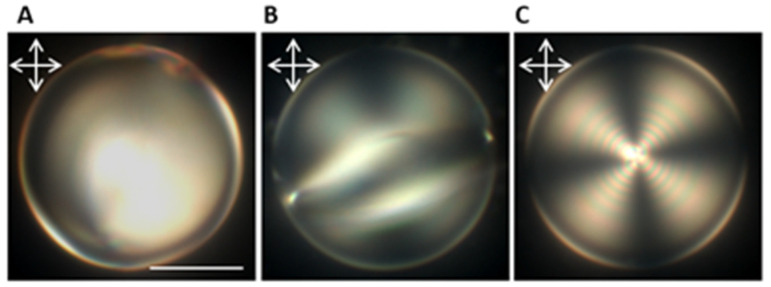
Reflection mode POM images of bipolar to radial ordering transition in a 50 µm nematic droplet in the presence of DLPC amphiphiles. It transient from bipolar (**A**) to intermediate (**B**) and finally radial configuration (**C**) (scale bar = 20 µm).

## Data Availability

All study data are included in the article and/or the Appendix A.

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
