# Peer review of "Chiral Liquid Crystal Microdroplets for Sensing Phospholipid Amphiphiles"

_biosensors, 2022, doi:10.3390/bios12050313_

Round 1
Reviewer 1 Report
The manuscript reports a study of interaction between chiral liquid crystal (CLC) microdroplets of low and high chirality and phospholipid amphiphiles in an aqueous environment. The CLC droplets with planar surface anchoring and diameters from 15 to 50 micrometers are produced in a microfluidic device, and exposed to different concentrations of DLPC phospholipid. The DLPC amphiphiles trigger structural reorganization and gradual transitions inside CLC droplets which can be observed with polarized light microscopy and UV-vis spectrometry. The authors study the effects of CLC droplet size and the related effect of curvature as well as the pitch modulation (the chirality level), the influence of DLPC concentration, and they finally compare the results with achiral nematic droplets. Their findings are sound, clearly presented, and sufficiently explained. I like their simple but quantitative approach to distinguish the DLPC concentrations effects through evolution of the colorimetric channels that nicely complement the polarized microscopy imaging. I can recommend the publication of the manuscript in Biosensors after addressing few minor comments below.
The introduction is informative and well-structured but the list of references should be supplemented by several recent works by other groups. The authors should cite the papers by Gleeson and coworkers (see Lab Chip 19, 1082 (2019), Langmuir 36, 6436 (2020), Mol. Syst. Des. Eng., doi:10.1039/d1me00189b (2022)), the preprint by Honaker et al. (doi:10.1101/2021.10.25.465736), a recent article by Lagerwall group (Phys. Rev. Research 4, 013130 (2022), and for instance Popov et al., Sens. Bio.-Sens. Res. 8, 31 (2016), and Ortiz et al. in ACS Appl. Mater. Interfaces 12, 29056 (2020). In my view, the authors focus on LC and CLC droplets only though there are many examples of surfactant and lipid triggered anchoring transitions in nematic and cholesteric shells too; see works by Lopez-Leon, Tran, and Lagerwall. The authors report that they have managed to precisely control the CLC droplet diameters in a microfluidic device but do not explain how the precision of less than 10 microns difference had been achieved. The evolution of structural patterns on droplet surface is complex and challenging to follow, however, it would be good to explain what is meant by threshold of DLPC molecules detection – should the micelles completely cover the CLC droplet surface or is it possible to observe only localized changes on surface area? In Fig. 3, time scale would be helpful. Is there any evidence of droplet rotation or dissolution under higher DLPC concentrations in the course of several hours? Nematic and cholesteric droplets of similar size have been also studied as microswimmers under different anchoring and surfactant conditions. Could a presence of external flow field accelerate and facilitate the lipid adsorption process?
Author Response
Dear Referee,
Thank you for your insightful comments and points. I uploaded the responses, pleases see the attached file.
Best Regards,
Sepideh Norouzi

Reviewer 2 Report
This is very well written manuscript on ordering transitions caused by DLPC on spherical chiral liquid crystal (CLC) droplets. Changes in the molecular reorientation of the CLC droplets in the presence of DLPC amphiphiles as a function of DLPC concentration were explored. To examine the effect of curvature, a microfluidic device was used to produce well-controlled monodisperse CLC droplets. The CLC-DLPC interfacial interaction was studied at both low and high chirality levels. The results demonstrated that the CLC-DLPC interface undergoes planar to homeotropic ordering transition through multistage molecular reorientations. The low chirality droplets detected the presence of DLPC amphiphiles three times faster than the high chirality counterparts. This is an important contribution to the field of biosensing and I think it will be of considerable interest to the readers of this journal. I recommend that the authors consider my questions/comments in a revision prior to publication.
1) It appears that the difference in high versus low chirality is achieved with a chemical dopant. Is the molecular mechanism for this effect well elucidated? If so, can a brief discussion of this be included? I am wondering how the DLPC chains are affected by the chiral dopant. Can this be discussed?
2) The use of DLPC as the only lipid in this study is warranted due to the focus of the study. I would like to know if the authors predict that the behavior observed in this study would be changed if fatty acid chains other than the short saturated fatty chains on DLPC were used? What about unsaturated chains or longer saturated chains? Also, would the results of the Bragg reflection be different if different lipids were used?
3) In the Conclusions, the following sentence is found: “Furthermore, we found that the chiral droplets undergo complete reconfiguration at significantly lower concentrations of DLPC amphiphiles than the nematic droplets.” This sentence confuses me. What is meant by “nematic droplets’ in this sentence? Do nematic droplets refer to high and low chirality or the achiral system studied later in the manuscript?
4) The material after the conclusions section on page 9 needs to be updated. It appears that the “Funding” and “Acknowledgments” are the only sections that have the correct information in them. The “Author Contributions” and “Supplemental Information” sections are very important and need to be filled out correctly. I don’t think that this work needs the IRB section. Anyway, please get these sections filled in correctly. I am sure this just didn’t get transferred into the final document correctly.
Author Response
Dear Referee,
Thank you for your insightful comments and points. Please find the attachment.
Best Regards,
Sepideh Norouzi
